# GRAPH U-NET

## ABSTRACT

We consider the problem of representation learning for graph data. Convolutional neural networks can naturally operate on images, but have significant challenges in dealing with graph data. Given images are special cases of graphs with nodes lie on 2D lattices, graph embedding tasks have a natural correspondence with image pixel-wise prediction tasks such as segmentation. While encoder-decoder architectures like U-Net have been successfully applied on many image pixel-wise prediction tasks, similar methods are lacking for graph data. This is due to the fact that pooling and up-sampling operations are not natural on graph data. To address these challenges, we propose novel graph pooling (gPool) and unpooling (gUnpool) operations in this work. The gPool layer adaptively selects some nodes to form a smaller graph based on their scalar projection values on a trainable projection vector. We further propose the gUnpool layer as the inverse operation of the gPool layer. The gUnpool layer restores the graph into its original structure using the position information of nodes selected in the corresponding gPool layer. Based on our proposed gPool and gUnpool layers, we develop an encoder-decoder model on graph, known as the graph U-Net. Our experimental results on node classification tasks demonstrate that our methods achieve consistently better performance than previous models.

## 1 INTRODUCTION

Convolutional neural networks (CNNs) (LeCun et al., 1998) have demonstrated great capability in various challenging artificial intelligence tasks, especially in fields of computer vision (He et al., 2017; Huang et al., 2017) and natural language processing (Vaswani et al., 2017; Bahdanau et al., 2015). One common property behind these tasks is that both images and texts have grid-like structures. Elements on feature maps have locality and order information, which enables the application of convolutional operations (Defferrard et al., 2016).

In practice, many real-world data can be naturally represented as graphs such as social and biological networks. Due to the great success of CNNs on grid-like data, applying them on graph data is particularly appealing. Recently, there have been many attempts to extend convolutions to graph data (GNNs) (Kipf & Welling, 2017; Veličković et al., 2017; Gao et al., 2018). One common use of convolutions on graphs is to compute node representations (Hamilton et al., 2017; Ying et al., 2018). With learned node representations, we can perform various tasks on graphs such as node classification and link prediction.

Images can be considered as special cases of graphs, in which nodes lie on regular 2D lattices. It is this special structure that enables the use of convolution and pooling operations on images. Based on this relationship, node classification and embedding tasks have a natural correspondence with pixel-wise prediction tasks such as image segmentation (Noh et al., 2015; Gao & Ji, 2017). In particular, both tasks aim to make predictions for each input unit, corresponding to a pixel on images or a node in graphs.

In the computer vision field, pixel-wise prediction tasks have achieved major advances recently. Encoder-decoder architectures like the U-Net (Ronneberger et al., 2015) are state-of-the-art methods for these tasks. It is thus highly interesting to develop U-Net-like architectures for graph data. In addition to convolutions, pooling and up-sampling operations are essential building blocks in these architectures. However, extending these operations to graph data is highly challenging. Unlike grid-

like data such as images and texts, nodes in graphs have no spatial locality and order information as required by regular pooling operations.

To bridge the above gap, we propose novel graph pooling (gPool) and unpooling (gUnpool) operations in this work. Based on these two operations, we propose U-Net-like architectures for graph data. The gPool operation samples some nodes to form a smaller graph based on their scalar projection values on a trainable projection vector. As an inverse operation of gPool, we propose a corresponding graph unpooling (gUnpool) operation, which restores the graph to its original structure with the help of locations of nodes selected in the corresponding gPool layer. Based on the gPool and gUnpool layers, we develop a graph U-Net, which allows high-level feature encoding and decoding for network embedding. Results on node classification tasks demonstrate the effectiveness of our proposed methods as compared to previous methods.

## 2 RELATED WORK

Recently, there has been a rich line of research on graph neural networks (Gilmer et al., 2017). Inspired by the first order graph Laplacian methods, Kipf & Welling (2017) proposed graph convolutional networks (GCNs), which achieved promising performance on graph node classification tasks. The layer-wise forward-propagation operation of GCNs is defined as:

$$X_{\ell+1} = \sigma(\hat{D}^{-\frac{1}{2}}\hat{A}\hat{D}^{-\frac{1}{2}}X_\ell W_\ell), \tag{1}$$

where $\hat{A} = A + I$ is used to add self-loops in the input adjacency matrix $A$, $X_\ell$ is the feature matrix of layer $\ell$. The GCN layer uses the diagonal node degree matrix $\hat{D}$ to normalize $\hat{A}$. $W_\ell$ is a trainable weight matrix that applies a linear transformation to feature vectors. GCNs essentially perform aggregation and transformation on node features without learning trainable filters. Hamilton et al. (2017) tried to sample a fixed number of neighboring nodes to keep the computational footprint consistent. Veličković et al. (2017) proposed to use attention mechanisms to enable different weights for neighboring nodes. Schlichtkrull et al. (2018) used relational graph convolutional networks for link prediction and entity classification. Some studies applied GNNs to graph classification tasks (Duvenaud et al., 2015; Dai et al., 2016; Zhang et al., 2018).

In addition to convolution, some studies tried to extend pooling operations to graphs. Defferrard et al. (2016) proposed to use binary tree indexing for graph coarsening, which fixes indices of nodes before applying 1-D pooling operations. Simonovsky & Komodakis (2017) used deterministic graph clustering algorithm to determine pooling patterns. Ying et al. (2018) used an assignment matrix to achieve pooling by assigning nodes to different clusters of the next layer.

## 3 GRAPH U-NET

In this section, we introduce the graph pooling (gPool) layer and graph unpooling (gUnpool) layer. Based on these two new layers, we develop the graph U-Net for node classification tasks.

### 3.1 GRAPH POOLING LAYER

Pooling layers play important roles in CNNs on grid-like data. They can reduce sizes of feature maps and enlarge receptive fields, thereby giving rise to better generalization and performance (Yu & Koltun, 2016). On grid-like data such as images, feature maps are partitioned into non-overlapping rectangles, on which non-linear down-sampling functions like maximum are applied. In addition to local pooling, global pooling layers (Zhao et al., 2015) perform down-sampling operations on all input units, thereby reducing each feature map to a single number. In contrast, $k$-max pooling layers (Blunsom et al., 2014) select the $k$-largest units out of each feature map.

However, we cannot directly apply these pooling operations to graphs. In particular, there is no locality information among nodes in graphs. Thus the partition operation is not applicable. The global pooling operation will reduce all nodes to one, which restricts the flexibility of networks. The $k$-max pooling operation outputs the $k$-largest units that may come from different nodes, resulting in inconsistency in the connectivity of selected nodes.

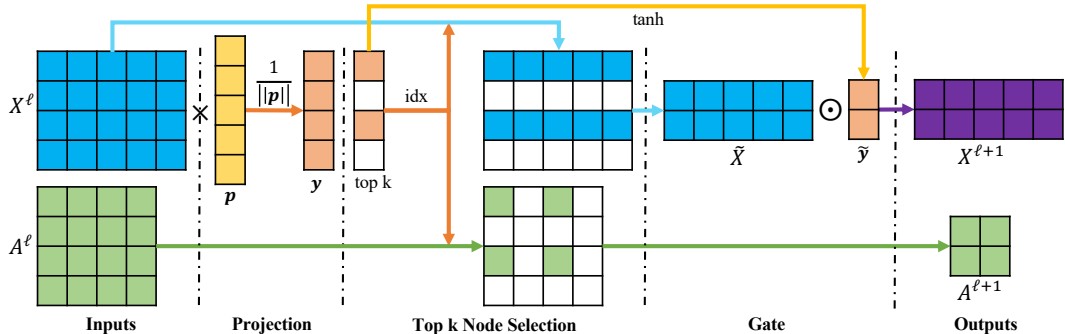

Figure 1: An illustration of the proposed graph pooling layer with $k = 2$. $\times$ and $\odot$ denote matrix multiplication and element-wise product, respectively. We consider a graph with 4 nodes, and each node has 5 features. By processing this graph, we obtain the adjacency matrix $A^\ell \in \mathbb{R}^{4\times4}$ and the input feature matrix $X^\ell \in \mathbb{R}^{4\times5}$ of layer $\ell$. In the projection stage, $\mathbf{p} \in \mathbb{R}^5$ is a trainable projection vector. By matrix multiplication and $\tanh(\cdot)$, we obtain $\mathbf{y}$ that are scores estimating scalar projection values of each node to the projection vector. By using $k = 2$, we select two nodes with the highest scores and record their indices in the top-k-node selection stage. We use the indices to extract the corresponding nodes to form a new graph, resulting in the pooled feature map $\tilde{X}^\ell$ and new corresponding adjacency matrix $A^{\ell+1}$. At the gate stage, we perform element-wise multiplication between $\tilde{X}^\ell$ and the selected node scores vector $\tilde{\mathbf{y}}$, resulting in $X^{\ell+1}$. This graph pooling layer outputs $A^{\ell+1}$ and $X^{\ell+1}$.

In this section, we propose the graph pooling (gPool) layer to enable down-sampling on graph data. In this layer, we adaptively select a subset of nodes to form a new but smaller graph. To this end, we employ a trainable projection vector $\mathbf{p}$. By projecting all node features to 1D, we can perform $k$-max pooling for node selection. Since the selection is based on 1D footprint of each node, the connectivity in the new graph is consistent across nodes. Given a node $i$ with its feature vector $\mathbf{x}_i$, the scalar projection of $\mathbf{x}_i$ on $\mathbf{p}$ is $y_i = \mathbf{x_i}\mathbf{p}/\|\mathbf{p}\|$. Here, $y_i$ measures how much information of node $i$ can be retained when projected onto the direction of $\mathbf{p}$. By sampling nodes, we wish to preserve as much information as possible from the original graph. To achieve this, we select nodes with the largest scalar projection values on $\mathbf{p}$ to form a new graph.

Suppose there are $N$ nodes in a graph $\mathbb{G}$ and each of which contains $C$ features. The graph can be represented by two matrices; those are the adjacency matrix $A^\ell \in \mathbb{R}^{N\times N}$ and the feature matrix $X^\ell \in \mathbb{R}^{N\times C}$. Row vector $\mathbf{x}_i^\ell$ in $X^\ell$ denotes the feature vector of node $i$ in the graph. The layer-wise propagation rule of graph pooling layer $\ell$ is defined as:

$$\mathbf{y} = X^\ell \mathbf{p}^\ell/\|\mathbf{p}^\ell\|, \qquad \text{idx} = \text{rank}(\mathbf{y}, k), \qquad \tilde{\mathbf{y}} = \tanh(\mathbf{y}(\text{idx})),$$
$$\tilde{X}^\ell = X^\ell(\text{idx}, :), \qquad A^{\ell+1} = A^\ell(\text{idx}, \text{idx}), \quad X^{\ell+1} = \tilde{X}^\ell \odot \left(\tilde{\mathbf{y}}\mathbf{1}_C^T\right), \tag{2}$$

where $k$ is the number of nodes selected in the new graph. $\text{rank}(\mathbf{y}, k)$ is the operation of node ranking, which returns indices of the $k$-largest values in $\mathbf{y}$. The idx returned by $\text{rank}(\mathbf{y}, k)$ contains the indices of nodes selected for the new graph. $A^\ell(\text{idx}, \text{idx})$ and $X^\ell(\text{idx}, :)$ perform the row and/or column extraction to form the adjacency matrix and the feature matrix for the new graph. $\mathbf{y}(\text{idx})$ extracts values in $\mathbf{y}$ with indices idx followed by a $\tanh$ operation. $\mathbf{1}_C \in \mathbb{R}^C$ is a vector of size $C$ with all components being 1, and $\odot$ represents the element-wise matrix multiplication.

$X^\ell$ is the feature matrix with row vectors $\mathbf{x}_1^\ell, \mathbf{x}_2^\ell, \cdots, \mathbf{x}_N^\ell$, each of which corresponds to a node in the graph. We first compute the scalar projection of $X^\ell$ on $\mathbf{p}^\ell$, resulting in $\mathbf{y} = [y_1, y_2, \cdots, y_N]^T$ with each $y_i$ measuring the scalar projection value of each node on the projection vector $\mathbf{p}^\ell$. Based on the scalar projection vector $\mathbf{y}$, $\text{rank}(\cdot)$ operation ranks values and returns the $k$-largest values in $\mathbf{y}$. Suppose the $k$-selected indices are $i_1, i_2, \cdots, i_k$ with $i_m < i_n$ and $1 \leq m < n \leq k$. Note that the index selection process preserves the position order information in the original graph. With indices idx, we extract the adjacency matrix $A^\ell \in \mathbb{R}^{k\times k}$ and the feature matrix $\tilde{X}^\ell \in \mathbb{R}^{k\times C}$ for the new graph. Finally, we employ a gate operation to control information flow. With selected indices idx, we obtain the gate vector $\tilde{y} \in \mathbb{R}^k$ by applying $\tanh$ to each element in the extracted scalar projection vector. Using element-wise matrix product of $\tilde{X}^\ell$ and $\tilde{\mathbf{y}}\mathbf{1}_C^T$, information of selected nodes

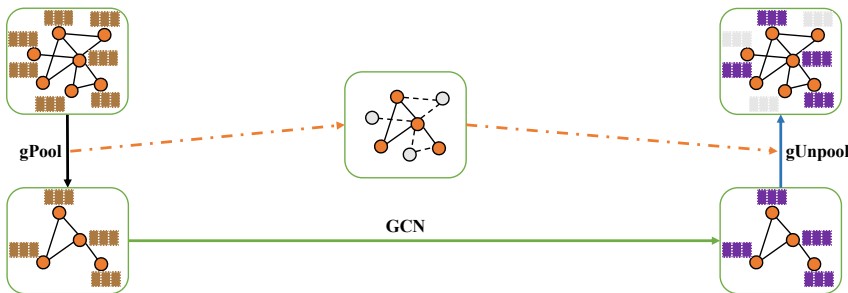

Figure 2: An illustration of the proposed graph unpooling (gUnpool) layer. In this example, a graph with 7 nodes is down-sampled using a gPool layer, resulting in a coarsened graph with 4 nodes and position information of selected nodes. The corresponding gUnpool layer uses the position information to reconstruct the original graph structure by using empty feature vectors for unselected nodes.

is controlled. The $i$th row vector in $X^{\ell+1}$ is the product of corresponding row vector in $X^\ell$ and the $i$th scalar value in $\tilde{y}$. Notably, the gate operation makes the projection vector $\mathbf{p}$ trainable by back-propagation (LeCun et al., 2012). Figure 1 provides an illustration of our proposed graph pooling layer. Compared to pooling operations used in grid-like data, our graph pooling layer employs extra training parameters in projection vector $\mathbf{p}$. We will show that the extra parameters are negligible but can boost performance.

## 3.2 GRAPH UNPOOLING LAYER

Up-sampling operations are important for encoder-decoder networks such as U-Net. The encoders of networks usually employ pooling operations to reduce feature map size and increase receptive field. While in decoders, feature maps need to be up-sampled to restore their original resolutions. On grid-like data like images, there are several up-sampling operations such as the deconvolution and unpooling layers. However, such operations are not currently available on graph data.

To enable up-sampling operations on graph data, we propose the graph unpooling (gUnpool) layer, which performs the inverse operation of the gPool layer and restores the graph into its original structure. To achieve this, we record the locations of nodes selected in the corresponding gPool layer and use this information to place nodes back to their original positions in the graph. Formally, we propose the layer-wise propagation rule of graph unpooling layer as

$$X^{\ell+1} = \text{distribute}(0_{N \times C}, X^\ell, \text{idx}), \tag{3}$$

where $\text{idx} \in \mathbb{Z}^{*k}$ contains indices of selected nodes in the corresponding gPool layer that reduces the graph size from $N$ nodes to $k$ nodes. $X^\ell \in \mathbb{R}^{k \times C}$ are the feature matrix of the current graph, and $0_{N \times C}$ are the initially empty feature matrix for the new graph. $\text{distribute}(0_{N \times C}, X^\ell, \text{idx})$ is the operation that distributes row vectors in $X^\ell$ into $0_{N \times C}$ feature matrix according to their corresponding indices stored in idx. In $X^{\ell+1}$, row vectors with indices in idx are updated by row vectors in $X^\ell$, while other row vectors remain zero.

## 3.3 GRAPH U-NET ARCHITECTURE

It is well-known that encoder-decoder networks like U-Net achieve promising performance on pixel-wise prediction tasks, since they can encode and decode high-level features while maintaining local spatial information. Similar to pixel-wise prediction tasks (Gong et al., 2014; Ronneberger et al., 2015), node classification tasks aim to make a prediction for each input unit. Based on our proposed gPool and gUnpool layers, we propose our graph U-Net (g-U-Net) architecture for node classification tasks.

In our graph U-Net (g-U-Net), we first apply a graph embedding layer to convert nodes into low-dimensional representations, since original inputs of some dataset like Cora (Sen et al., 2008) usually have very high-dimensional feature vectors. After the graph embedding layer, we build the encoder by stacking several encoding blocks, each of which contains a gPool layer followed by a GCN

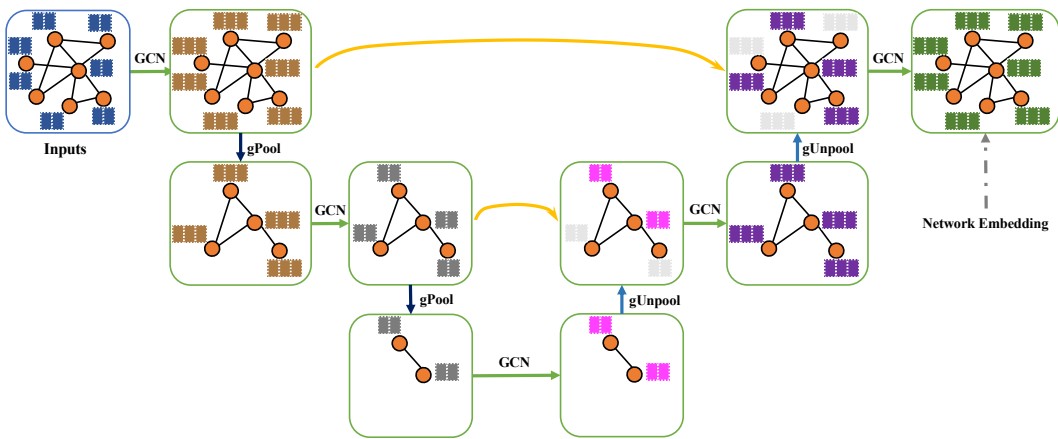

Figure 3: An illustration of the proposed graph U-Net (g-U-Net). In this example, each node in the input graph has two features. The input feature vectors are transformed into low-dimensional representations using a GCN layer. After that, we stack two encoder blocks, each of which contains a gPool layer and a GCN layer. In the decoder part, there are also two decoder blocks. Each block consists of a gUnpool layer and a GCN layer. For blocks in the same level, encoder block uses skip connection to fuse the low-level spatial features from the encoder block. The output feature vectors of nodes in the last layer are network embedding, which can be used for various tasks such as node classification and link prediction.

layer. gPool layers reduce the size of graph to encode higher-order features, while GCN layers are responsible for aggregating information from each node's first-order information. In the decoder part, we stack the same number of decoding blocks as in the encoder part. Each decoder block is composed of a gUnpool layer and a GCN layer. The gUnpool layer restores the graph into its higher resolution structure, and the GCN layer aggregates information from the neighborhood. There are skip-connections between corresponding blocks of encoder and decoder layers, which transmit spatial information to decoders for better performance. The skip-connection can be either feature map addition or concatenation. Finally, we employ a GCN layer for final predictions before the softmax function. Figure 3 provides an illustration of a sample g-U-Net with two blocks in encoder and decoder.

### 3.4 GRAPH CONNECTIVITY AUGMENTATION VIA GRAPH POWER

In our proposed gPool layer, we sample some important nodes to form a new graph for high-level feature encoding. Since related edges are removed when removing nodes in gPool, the nodes in the pooled graph might become isolated. This may influence the information propagation in subsequent layers, especially when GCN layers are used to aggregate information from neighboring nodes. We need to increase connectivity among nodes in the pooled graph. To address this problem, we propose to use the $k^{th}$ graph power $\mathbb{G}^k$ to increase the graph connectivity. This operation builds links between nodes whose distances are at most $k$ hops (Chepuri & Leus, 2016). In this work, we employ $k = 2$ since there is a GCN layer before each gPool layer to aggregate information from its first-order neighboring nodes. Formally, we replace the fifth equation in Eq 2 by:

$$A^2 = A^\ell A^\ell, \quad A^{\ell+1} = A^2(\mathrm{idx}, \mathrm{idx}), \tag{4}$$

where $A^2 \in \mathbb{R}^{N \times N}$ is the $2^{nd}$ graph power. Now, the graph sampling is performed on the augmented graph with better connectivity.

### 3.5 IMPROVED GCN LAYER

In Eq. 1, the adjacency matrix before normalization is computed as $\hat{A} = A + I$ in which a self-loop is added to each node in the graph. When performing information aggregation, the same weight is given to node's own feature vector and its neighboring nodes. In this work, we wish to give a higher weight to node's own feature vector, since its own feature should be more important for prediction.

Table 1: Summary of datasets used in our experiments (Yang et al., 2016; Zitnik & Leskovec, 2017). The Cora, Citeseer, and Pubmed datasets are used for transductive learning experiments.

| Dataset | Nodes | Features | Classes | Training | Validation | Testing | Degree |
|---------|-------|----------|---------|----------|------------|---------|--------|
| Cora | 2708 | 1433 | 7 | 140 | 500 | 1000 | 4 |
| Citeseer | 3327 | 3703 | 6 | 120 | 500 | 1000 | 5 |
| Pubmed | 19717 | 500 | 3 | 60 | 500 | 1000 | 6 |

Table 2: Results of transductive learning experiments in terms of node classification accuracies on Cora, Citeseer, and Pubmed datasets. g-U-Net denotes our proposed graph U-Net model.

| Models | Cora | Citeseer | Pubmed |
|--------|------|----------|--------|
| DeepWalk (Perozzi et al., 2014) | 67.2% | 43.2% | 65.3% |
| Planetoid (Yang et al., 2016) | 75.7% | 64.7% | 77.2% |
| Chebyshev (Defferrard et al., 2016) | 81.2% | 69.8% | 74.4% |
| GCN (Kipf & Welling, 2017) | 81.5% | 70.3% | 79.0% |
| GAT (Veličković et al., 2017) | $83.0 \pm 0.7\%$ | $72.5 \pm 0.7\%$ | $79.0 \pm 0.3\%$ |
| **g-U-Net (Ours)** | **$84.4 \pm 0.6\%$** | **$73.2 \pm 0.5\%$** | **$79.6 \pm 0.2\%$** |

To this end, we change the calculation to $\hat{A} = A + 2I$ by imposing larger weights on self loops in the graph, which is common in graph processing. All experiments in this work use this modified version of GCN layer for better performance.

# 4 EXPERIMENTAL STUDY

In this section, we evaluate our gPool and gUnpool layers based on the g-U-Net proposed in Section 3.3. We compare our networks with previous state-of-the-art models on node classification tasks. Experimental results show that our methods achieve new state-of-the-art results in terms of node classification accuracy. Some ablation studies are performed to examine the contributions of the proposed gPool layer, gUnpool layer, and graph connectivity augmentation to performance improvements. We conduct studies on the relationship between network depth and node classification performance. We investigate if additional parameters involved in gPool layers can increase the risk of over-fitting.

## 4.1 DATASETS

In experiments, we evaluate our networks on node classification tasks under transductive learning settings. Under this setting, unlabeled data are accessible for training, which enables the network to learn about the graph structure. To be specific, only part of nodes are labeled while labels of other nodes in the same graph remain unknown. We employ three benchmark datasets for this setting; those are Cora, Citeseer, and Pubmed (Kipf & Welling, 2017), which are summarized in Table 1. These datasets are citation networks, with each node and each edge representing a document and a citation, respectively. The feature vector of each node is the bag-of-word representation whose dimension is determined by the dictionary size. We follow the same experimental settings in (Kipf & Welling, 2017). For each class, there are 20 nodes for training, 500 nodes for validation, and 1000 nodes for testing.

## 4.2 EXPERIMENTAL SETUP

For transductive learning tasks, we employ our proposed g-U-Net proposed in Section 3.3. Since nodes in the three datasets are associated with high-dimensional features, we employ a GCN layer to reduce them into low-dimensional representations. In the encoder part, we stack four blocks, each of which consists of a gPool layer and a GCN layer. We sample 2000, 1000, 500, 200 nodes in the four gPool layers, respectively. Correspondingly, the decoder part also contains four blocks. Each decoder block is composed of a gUnpool layer and a GCN layer. We use addition operation in skip connections between blocks of encoder and decoder parts. Finally, we apply a GCN layer for final

Table 3: Comparison of g-U-Nets with and without gPool or gUnpool layers in terms of node classification accuracy on Cora, Citeseer, and Pubmed datasets.

| Models | Cora | Citeseer | Pubmed |
|---|---|---|---|
| g-U-Net without gPool or gUnpool | $82.1 \pm 0.6\%$ | $71.6 \pm 0.5\%$ | $79.1 \pm 0.2\%$ |
| **g-U-Net (Ours)** | **$84.4 \pm 0.6\%$** | **$73.2 \pm 0.5\%$** | **$79.6 \pm 0.2\%$** |

Table 4: Comparison of g-U-Nets with and without graph connectivity augmentation in terms of node classification accuracy on Cora, Citeseer, and Pubmed datasets.

| Models | Cora | Citeseer | Pubmed |
|---|---|---|---|
| g-U-Net without augmentation | $83.7 \pm 0.7\%$ | $72.5 \pm 0.6\%$ | $79.0 \pm 0.3\%$ |
| **g-U-Net (Ours)** | **$84.4 \pm 0.6\%$** | **$73.2 \pm 0.5\%$** | **$79.6 \pm 0.2\%$** |

prediction. For all layers in the model, we use identity activation function. To avoid over-fitting, we apply $L_2$ regularization on weights with $\lambda = 0.001$. Dropout (Srivastava et al., 2014) is applied to both adjacency matrices and feature matrices with rates of 0.8 and 0.08, respectively.

## 4.3 PERFORMANCE STUDY

Under transductive learning settings, we compare our proposed g-U-Net with other state-of-the-art models in terms of node classification accuracy. We report node classification accuracies on datasets Cora, Citeseer, and Pubmed, and the results are summarized in Table 2. We can observe from the results that our g-U-Net achieves consistently better performance than other networks. For baseline values listed for node classification tasks, they are the state-of-the-art on these datasets. Our proposed model is composed of GCN, gPool, and gUnpool layers without involving more advanced graph convolution layers like GAT. When compared to GCN directly, our g-U-Net significantly improves performance on all three datasets by margins of 2.9%, 2.9%, and 0.6%, respectively. Note that the only difference between our g-U-Net and GCN is the use of encoder-decoder architecture containing gPool and gUnpool layers. These results demonstrate the effectiveness of g-U-Net in network embedding.

## 4.4 ABLATION STUDY OF GPOOL AND GUNPOOL LAYERS

Although GCNs have been reported to have worse performance when the network goes deeper (Kipf & Welling, 2017), it may also be argued that the performance improvement over GCN in Table 2 is due to the use of a deeper network architecture. In this section, we investigate the contributions of gPool and gUnpool layers to the performance of g-U-Net. We conduct experiments by removing all gPool and gUnpool layers from our g-U-Net, leading to a network with only GCN layers with skip connections. Table 3 provides the comparison results between g-U-Nets with and without gPool or gUnpool layers. The results show that g-U-Net has better performance over g-U-Net without gPool or gUnpool layers on all three datasets. These results demonstrate the contributions of gPool and gUnpool layers to performance improvement. When considering the difference between the two models in terms of architecture, g-U-Net enables higher level feature encoding, thereby resulting in better generalization and performance.

## 4.5 GRAPH CONNECTIVITY AUGMENTATION STUDY

In the above experiments, we employ gPool layers with graph connectivity augmentation by using the $2^{nd}$ graph power in Section 3.4. Here, we conduct experiments to investigate the benefits of graph connectivity augmentation based on g-U-Net. We remove the graph connectivity augmentation from gPool layers while keeping other settings the same for fairness of comparisons. Table 4 provides comparison results between g-U-Nets with and without graph connectivity augmentation. The results show that the absence of graph connectivity augmentation will cause consistent performance degradation on all of three datasets. This demonstrates that graph connectivity augmentation via $2^{nd}$ graph power can help with information transfer among nodes in sampled graphs.

Table 5: Comparison of different network depths in terms of node classification accuracy on Cora, Citeseer, and Pubmed datasets. Based on g-U-Net, we try different network depths in terms of the number of blocks in encoder and decoder parts.

| Depth | Cora | Citeseer | Pubmed |
|-------|------|----------|--------|
| 2 | $82.6 \pm 0.6\%$ | $71.8 \pm 0.5\%$ | $79.1 \pm 0.3\%$ |
| 3 | $83.8 \pm 0.7\%$ | $72.7 \pm 0.7\%$ | $79.4 \pm 0.4\%$ |
| 4 | $\mathbf{84.4 \pm 0.6\%}$ | $\mathbf{73.2 \pm 0.5\%}$ | $\mathbf{79.6 \pm 0.2\%}$ |
| 5 | $84.1 \pm 0.5\%$ | $72.8 \pm 0.6\%$ | $79.5 \pm 0.3\%$ |

Table 6: Comparison of the g-U-Net with and without gPool or gUnpool layers in terms of the node classification accuracy and the number of parameters.

| Models | Accuracy | #Params | Ratio of increase |
|--------|----------|---------|-------------------|
| g-U-Net without gPool or gUnpool | $82.1 \pm 0.6\%$ | 75,643 | 0.00% |
| **g-U-Net (Ours)** | $\mathbf{84.4 \pm 0.6\%}$ | 75,737 | 0.12% |

## 4.6 NETWORK DEPTH STUDY OF GRAPH U-NET

Since the network depth in terms of the number of blocks in encoder and decoder parts is an important hyper-parameter in the g-U-Net, we conduct experiments to investigate the relationship between network depth and performance in terms of node classification accuracy. The results are summarized in Table 5. From the results, we can observe that the performance improves as network goes deeper until the depth of 4. The over-fitting problem prevents the network to improve when the depth goes beyond that. In the field of image segmentation, U-Net models with depth 3 or 4 are commonly used, which is consistent with our choice in experiments. This indicates the capacity of gPool and gUnpool layers in receptive field enlargement and high-level feature encoding even working with shallow networks.

## 4.7 PARAMETER STUDY OF GRAPH POOLING LAYERS

Since our proposed gPool layer involves extra parameters, we compute the number of additional parameters based on our g-U-Net. The comparison results between g-U-Net with and without gPool or gUnpool layers on dataset Cora are summarized in Table 6. From the results, we can observe that gPool layers in U-Net model only adds 0.12% additional parameters but can promote the performance by a margin of 2.3%. We believe this negligible increase of extra parameters will not increase the risk of over-fitting. Compared to g-U-Net without gPool or gUnpool layers, the encoder-decoder architecture with our gPool and gUnpool layers yields significant performance improvement.

## 5 CONCLUSION

In this work, we propose novel gPool and gUnpool layers in g-U-Net networks for network embedding. The gPool layer implements the regular global $k$-max pooling operation on graph data. It samples a subset of important nodes to enable high-level feature encoding and receptive field enlargement. By employing a trainable projection vector, gPool layers sample nodes based on their scalar projection values. Furthermore, we propose the gUnpool layer which applies unpooling operations on graph data. By using the position information of nodes in the original graph, gUnpool layer performs the inverse operation of the corresponding gPool layer and restores the original graph structure. Based on our gPool and gUnpool layers, we propose the graph U-Net (g-U-Net) architecture which uses a similar encoder-decoder architecture as regular U-Net on image data. Experimental results demonstrate that our g-U-Net achieves performance improvements as compared to other GNNs on transductive learning tasks. To avoid the isolated node problem that may exist in sampled graphs, we employ the $2^{nd}$ graph power to improve graph connectivity. Ablation studies indicate the contributions of our graph connectivity augmentation approach.

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
