# OpenReview forum: "Graph U-Net"
_ICLR.cc/2019/Conference_

### Official Review · AnonReviewer2 · 2018-10-29
**Good paper, clearly written and has some interesting ideas**

**Rating:** 7
**Confidence:** 4

**Review:**

Summary:
This paper introduces an encoder-decoder neural net architecture for arbitrary graphs. The core contribution is pooling and un-pooling operations for respectively graph down and up sampling.

Pros:
+ U-Net like architectures indeed are very successful in vision applications, and having a model that was similar properties on graphs would be very useful.
+ The paper is clearly written.
+ I really liked the idea behind the pooling operation: it is simple, seems easy to implement efficiently, and generally makes sense (although see concerns below).
+ The choice of the baselines is reasonable, and experimental results seem convincing. Ablation studies are also there.

Cons:
- It is not clear why the evaluation seem to only be done for the transductive learning settings. I understand that some of the previous work might have done that, but this application scenario is quite limited.
- One concern about the g-pool operation is that it is not local: unlike e.g. max pool on 2D which produces local maxima, here the selection is done globally, which could lead to situations where the entire parts of the graph are completely ignored.
- Another concern, which has been partially addressed in section 3.4 is that the connectivity is not really taken into account when downsampling the adjacency matrix. The solution which introduces previously non-existing edges and thus kind of modifies the original graph is not very satisfying.

---

> ### Author Response · Authors · 2018-11-21
> **Rebuttal to AnonReviewer2**
>
> Thank you for your comments.
>
> "- It is not clear why the evaluation seem to only be done for the transductive learning settings. I understand that some of the previous work might have done that, but this application scenario is quite limited."
>
> Thank you for your suggestions. We applied our proposed approaches to graph classification problems under inductive learning settings. We add more experiments on several graph classification datasets including D&D, Proteins, and Collab datasets, which are standard datasets employed in graph classification tasks under inductive learning settings. The results are summarized in Table below. It can be seen that our proposed pooling method outperforms DiffPool [1] by margins of 1.79% and 1.43% on two datasets.  Notably, the result reported by DiffPool-DET on Collab is significantly higher than other baselines and the other two DiffPool models. This demonstrates that our proposed methods can be applied to node classification and graph classification tasks under both transductive and inductive learning settings.
>
> ++++++++++++++++++++++++++++++++++++++++++++++
> ___________________|____D&D__|__ PROTEINS__|__COLLAB__
> ________PSCN_____|___ 76.27__|____75.00_____ |____72.60____
> _______DGCNN___|___ 79.37__|____76.26_____ |____73.76____
> ___DiffPool-DET_|___ 75.47__|____75.62_____ |____82.13____
> _DiffPool-NOLP_|___ 79.98__|____76.22_____ |____75.58____
> ______DiffPool___|___ 80.64__|____76.25_____ |____75.48____
> ______g-U-Net___|___ 82.43__|____77.68_____ |____77.56____
>
> Due to time limitation, we will add these results in the final version of our paper.
>
> "- One concern about the g-pool operation is that it is not local: unlike e.g. max pool on 2D which produces local maxima, here the selection is done globally, which could lead to situations where the entire parts of the graph are completely ignored. "
>
> Yes, our gPool operation is performed on global scope instead of local. This is due to the fact that defining locality is very hard especially for pooling operations. Unlike grid-like data such as images and texts, there is no obvious rule to group some nodes into a local patch for pooling operations. In DiffPool [1], they also define a global pooling operation. The difference is that their method learns an assignment matrix to softly assign each node to nodes in the new graph. While our approach is more similar to the regular global k-max pooling operation.
>
> Although some parts of the graph are abandoned in a gPool layer, our proposed gUnpool layer and graph U-Net architecture will restore the graph structure in the decoder part for feature representation learning. Therefore, we don’t need to worry about the loss of node information by employing gUnpool layer and graph U-Net architecture.
>
> "- Another concern, which has been partially addressed in section 3.4 is that the connectivity is not really taken into account when downsampling the adjacency matrix. The solution which introduces previously non-existing edges and thus kind of modifies the original graph is not very satisfying. "
>
> When performing pooling operation, the non-existing edges are introduced based on the fact that we employ GCN layers before our proposed gPool layers. Each GCN layer will aggregate one-hop neighboring nodes information for each node in the graph. This means that two nodes that are two hops away will have information communication. Based on this fact, we employ graph power of 2 to augment the graph connectivity to avoid isolated nodes in the graph.
>
> Also, this method will partially solve the connectivity loss problem when down-sampling the graph. But it’s hard to maintain original graph connectivity when we need to sample some important nodes out especially on sparsely-connected graphs.
>
> [1] Rex Ying, Jiaxuan You, Christopher Morris, Xiang Ren, William L Hamilton, and Jure
> Leskovec. Hierarchical graph representation learning with differentiable pooling. The Thirty-second Annual Conference on Neural Information Processing Systems (NIPS), 2018

---

> > ### Comment · AnonReviewer2 · 2018-11-26
> > **Re: rebuttal**
> >
> > Thank you for the response. I stand by my original rating.

---

### Official Review · AnonReviewer1 · 2018-11-02
**interesting problem of pooling/upsampling graphs, experimental validation and literature review could be significantly improved**

**Rating:** 4
**Confidence:** 4

**Review:**

This paper proposes pooling and upsampling operations for graph structured data, to be interleaved with graph convolutions, following the spirit of fully convolutional networks for image pixel-wise prediction. Experiments are performed on node classification benchmarks, showing an improvement w.r.t. architectures that do not perform any downsampling/upsampling operations.

Given that the main contribution of the paper is the introduction of a pooling operation for graph structured data, it might be a good idea to evaluate the operation in a task that does require some kind of downsampling, such as graph classification / regression. Moreover, authors should compare to other graph pooling methods.

Authors claim that one of the motivations to perform their pooling operation is to increase the receptive field. It would be worth comparing pooling/upsamping to dilated convolutions to see if they have the same effect on the performance when dealing with graphs.

Some choices in the method seem rather arbitrary, such as the tanh non-linearity in \tilde y. Could the authors elaborate on that? How important is the gating?

It would be interesting to analyze which nodes where selected by the pooling operators. Are those nodes close together or spread out in the previous graph?

The proposed unpooling operation seems to be the same as unpooling performed to upsample images, that is using skip connections to track indices, by recovering the position where the max value comes from and setting the rest to 0. Have the authors tried other upsampling strategies analogous to the ones typically used for images (e.g. upsampling with nearest neighbors)?

When skipping information from the downsampling path to the upsampling path, is there a concatenation or a summation? How do both operations compare? (note that concatenation introduces many more parameters) How about only skipping only the indices (no summation nor concatenation)? This kind of analysis, as it has been done in the computer vision literature, would be interesting.

What is the influence of the first embedding layer to reduce the dimensionality of the features?

How do the models in Table 2 compare in terms of number of parameters?
 What's the influence of imposing larger weights on self loop in the graph?

What about experiments in inductive settings?

Please add references for the following claim "U-Net models with depth 3 or 4 are commonly used..."

Please double check your references, e.g. in the introduction, citations used for CNNs do not always correspond to CNN architectures.

The literature review could be significantly improved, missing relevant papers to discuss include:
- Gori et al. A new model for learning in graph domains, 2005.
- Scarselli et al. The graph neural network model, 2009.
- Bruna et al. Spectral networks and locally connected networks on graphs, 2014.
- Henaff et al. Deep convolutional networks on graph-structured data, 2015.
- Niepert et al. Learning convolutional neural networks for graphs, 2016.
- Atwood and Towsley. Diffusion-convolutional neural networks, 2016.
- Bronstein et al. Geometric deep learning: going beyond Euclidean data, 2016.
- Monti et al. Geometric deep learning on graphs and manifolds using mixture model cnns, 2017.
- Fey et al. SplineCNN: Fast Geometric Deep Learning with Continuous B-Spline Kernels, 2017.
- Gama et al. Convolutional Neural Networks Architectures for Signals Supported on Graphs, 2018.
As well as other pixel-wise architecture for image-based tasks such as:
- Long et al. Fully Convolutional Networks for Semantic Segmentation, 2015.
- Jegou et al. The one hundred layers tiramisu: fully convolutional densenets for semantic segmentation, 2016.
- Isola et al. Image-to-image translation with conditional adversarial networks, 2016.
- Zhao et al. Stacked What-Where auto-encoders, 2015.

---

> ### Author Response · Authors · 2018-11-21
> **Rebuttal to AnonReviewer1 (Part 2)**
>
> Thank you for your comments.
>
> "-Some choices in the method seem rather arbitrary, such as the tanh non-linearity in \tilde y. Could the authors elaborate on that? How important is the gating?"
>
> The choices are not arbitrary. In our experiments, the tanh outperforms other commonly employed gated operations. As we stated in Section 3.1 of our paper, the gated operations are very important for training the projection vector p: “Notably, the gate operation makes the projection vector p trainable by backpropagation…”.
>
> "-It would be interesting to analyze which nodes where selected by the pooling operators. Are those nodes close together or spread out in the previous graph?"
>
> Thank you for your suggestion. Due to time limitation, we will add some graph visualization in the final version of our paper.
>
> "-...Have the authors tried other upsampling strategies analogous to the ones typically used for images (e.g. upsampling with nearest neighbors)?"
>
> We cannot try other up-sampling strategies since there are still no up-sampling methods for graphs currently. Most up-sampling operations on images need locality information such as deconvolution layer. But it is hard on graph data since numbers of neighing nodes are not fixed and they are not ordered.
>
> "-When skipping information from the downsampling path to the upsampling path, is there a concatenation or a summation? How do both operations compare? (note that concatenation introduces many more parameters) How about only skipping only the indices (no summation nor concatenation)? This kind of analysis, as it has been done in the computer vision literature, would be interesting."
>
> We use summation for skip connection. We compared these two skip connection strategies and found summation worked better. Summation operation can reduce the number of parameters compared to concatenation, which helps to avoid overfitting. Due to page limit, and these are not our main contribution, we did not put such information on the paper. When published, we will release our code, which includes all such implementation details.
>
> "-What is the influence of the first embedding layer to reduce the dimensionality of the features?"
>
> Since we worked on three citation network datasets, the initial feature vectors are bag-of-words representations, which is really sparse and has high-dimension. We use an embedding layer to reduce them into low-dimensional representations to avoid over-fitting. In practice, this layer is very important since it helps reduce parameters, thereby resulting in better generalization and performance.
>
> "-How do the models in Table 2 compare in terms of number of parameters?"
>
> We didn’t provide such kind of comparisons in our paper since baseline models did not report the number of parameters in their works. But in our models, the numbers of parameters are only around 20 thousand depending on the datasets.
>
> "-What's the influence of imposing larger weights on self loop in the graph?"
>
> Imposing larger weights promotes the performance slightly. Since it is a very popular way in traditional machine learning methods on graph data, we did not provide analysis on this.
>
> "-What about experiments in inductive settings?"
>
> We add experiments on graph classification tasks which are under inductive learning settings. The results are summarized in Table above. Our approaches achieve new state-of-the-art performance on graph classification tasks under inductive settings.
>
> "-Please add references for the following claim "U-Net models with depth 3 or 4 are commonly used...""
>
> Sure, we will add some. Thanks.
>
> "-Please double check your references, e.g. in the introduction, citations used for CNNs do not always correspond to CNN architectures. "
>
> Sure, we will. Thanks.
>
> "-The literature review could be significantly improved, missing relevant papers to discuss include:..."
>
> Thank you for providing these helpful references. I will add some related to the paper. But some references listed here are not very related to my work. For example, - Bruna et al. Spectral networks and locally connected networks on graphs, 2014 works on spectral networks. Also, Isola et al. Image-to-image translation with conditional adversarial networks, 2016 is a GAN work on images and is not related to my work. I will add some related works in the final version of our paper. Thanks.

---

> ### Author Response · Authors · 2018-11-21
> **Rebuttal to AnonReviewer1 (Part 1)**
>
> Thank you for your comments.
>
> "-Given that the main contribution of the paper is the introduction of a pooling operation for graph structured data, it might be a good idea to evaluate the operation in a task that does require some kind of downsampling..."
>
> To evaluate our gPool layer on down-sampling-required tasks, we add more experiments on graph classification tasks under inductive learning settings on three standard datasets; those are D&D, Proteins, and Collab datasets with 1178, 1113, and 5000 graphs, respectively. The results including comparison with a state-of-the-art graph pooling method [1] are summarized in Table below. Our proposed methods outperform baseline models including DiffPool on two out of three datasets and achieve new state-of-the-art performances.  Notably, the result reported by DiffPool-DET on Collab is significantly higher than other baselines and the other two DiffPool models.
>
> Note that the primary contribution of our work is to develop both graph pooling and unpooling layers that together enable the development of graph U-nets. Evaluation of our methods on graph classification tasks only involve pooling layers, which is not a comprehensive evaluation of our proposed methods.
>
> ++++++++++++++++++++++++++++++++++++++++++++++
> __________________|____D&D__|__ PROTEINS__|__COLLAB__
> ________PSCN____|___ 76.27__|____75.00_____|____72.60____
> _______DGCNN__ |___ 79.37__|____76.26_____|____73.76____
> ___DiffPool-DET_|___ 75.47__|____75.62_____|____82.13____
> _DiffPool-NOLP_|___ 79.98__|____76.22_____|____75.58____
> ______DiffPool___|___ 80.64__|____76.25_____|____75.48____
> ______g-U-Net___|___ 82.43__|____77.68_____|____77.56____
>
> Due to time constraint, we will add these results in the final version of our paper.
>
> "-Authors claim that one of the motivations to perform their pooling operation is to increase the receptive field. It would be worth comparing pooling/upsamping to dilated convolutions..."
>
> Thanks for your suggestion. Dilated convolution is not defined on graphs since it is not clear how to define locality on graph data. Actually, regular convolution operations are not available on graph data. GCN only performs a linear transformation after a simple summation from neighboring nodes. To our knowledge, trainable filters on spatial dimension are not available on graph data. It’s hard to compare with dilated convolutions on graph data.
>
> [1] Rex Ying, Jiaxuan You, Christopher Morris, Xiang Ren, William L Hamilton, and Jure
> Leskovec. Hierarchical graph representation learning with differentiable pooling. The Thirty-second Annual Conference on Neural Information Processing Systems (NIPS), 2018

---

### Official Review · AnonReviewer3 · 2018-11-02
**An interesting paper that could benefit from more empirical comparisons**

**Rating:** 7
**Confidence:** 5

**Review:**

* I have revised my score upwards due to the authors response to my concerns --- particularly the addition of new results on graph classification. The original review remains here, and I respond to the author's response below.

The authors propose a new technique to add “pooling” and “unpooling” layers to a graph neural network (GNN). To deal with the lack of spatial locality in graphs, the downsampling operation relies on a learned scalar projection vector (which gives the “scores” for selecting different nodes). During upsampling, the model simple relies on storing the un-sampled adjacency matrix. Thorough experimental results on Cora, Citeseer, and Pubmed highlight the utility of the approach, with ablation studies isolating the importance of the pool/unpool operations.

Overall, this is an interesting paper with the possibility of having a moderate impact within the area of GNNs/GCNs, and the method is clearly described. While there are a number of minor modifications made to the standard GCN model, which could potentially confound the results, the authors do provide a sensible ablation study to isolate the importance of their pool/unpool operations. The overall results on the three node classification datasets are also quite strong.

The primary shortcoming of this paper is that it only evaluates the model on three citation network datasets (Cora, Citseer, and Pubmed). While these datasets are now standard in the GCN/GNN community, they are very small, have few labeled examples, and it would greatly strengthen the paper to use a different dataset or two, e.g., the Reddit or PPI datasets from Hamilton et al. 2017 or the BlogCatolog dataset used in Grover et al. 2016 could be used for node classification. Or the authors could apply the proposed technique to graph classification or link prediction. In this reviewers opinion, it is very hard to judge the general utility of a method when results are only provided on these three very-specific datasets, where the performance differences between methods are now very marginal.

In a related point, while this work cites other approaches that apply pooling operations in graph neural networks (e.g., Ying et al. 2018, Simonovsky and Komodakis 2018), no comparisons are made against these approaches. One would suppose that these comparisons are not made because this paper only tests the graph U-net for node classification, but it would greatly strengthen this paper to add comparisons to these other pooling operations, e.g., for graph classification. Moreover, it is possible to define analogous unpooling operations for Ying et al. 2018 and Simonovsky and Komodakis 2018, similar to the unpooling operation used in this work (e.g., for Ying et al.’s DiffPool you can just “unpool” to the previous graph and assign each node a feature corresponding to the weighted sum of the features of the assigned clusters). Of course, it would require significant work (e.g., experiments on graph classification or some modifications of existing approaches) to actually test whether the pool approach proposed here is actually better than those in Ying et al. 2018 and Simonovsky and Komodakis 2018, but such comparisons are necessary to demonstrate whether the pooling operation proposed here is an improvement over existing works, or whether the primary novelty is the combined application of pooling and unpooling in a node classification setting.

As another minor point, whereas unpooling operations can be used to define a generative model in the image setting, this is not the case here, as the unpooling operation relies on knowledge about the input graph (i.e., the model always unpools to the same connectivity structure). This is not necessarily a bad thing, but it could improve the paper to clarify this issue.

---

> ### Author Response · Authors · 2018-11-21
> **Rebuttal to AnonReviewer3**
>
> Thanks for your comments.
>
> "-The primary shortcoming of this paper is that it only evaluates the model on three citation network datasets (Cora, Citseer, and Pubmed). While these datasets are now standard in the GCN/GNN community, they are very small, have few labeled examples, and it would greatly strengthen the paper to use a different dataset or two, e.g., the Reddit or PPI datasets from Hamilton et al. 2017 or the BlogCatolog dataset used in Grover et al. 2016 could be used for node classification. Or the authors could apply the proposed technique to graph classification or link prediction. In this reviewers opinion, it is very hard to judge the general utility of a method when results are only provided on these three very-specific datasets, where the performance differences between methods are now very marginal. "
>
> To evaluate our proposed gPool method for graph classification tasks on other datasets, we add more experiments on several graph classification datasets, including D&D, Proteins, and Collab datasets, which are standard datasets employed in such experiment settings. D&D, Proteins, and Collab datasets contain 1178, 1113, and 5000 graphs and 284.32, 39.06, and 74.49 average numbers of nodes on each graph, respectively. The results are summarized in Table below. We can observe from the results that our proposed gPool method outperforms DiffPool [1] by margins of 1.79% and 1.43% on D&D and Proteins. Notably, the result obtained by DiffPool-DET on Collab is significantly higher than all other methods and the other two DiffPool models. On all three datasets, our model outperforms baseline models including DiffPool. In addition, DiffPool claimed that their training utilized auxiliary task of link prediction to stabilize model performance. But in our experiments, we only use graph labels for training without any auxiliary tasks to stabilize training.
>
> ++++++++++++++++++++++++++++++++++++++++++++++
> ___________________|____D&D__|__ PROTEINS__|__COLLAB__
> ________PSCN_____|___ 76.27__|____75.00_____|____72.60____
> ______DGCNN____|___ 79.37__|____76.26_____|____73.76____
> __ DiffPool -DET_|___ 75.47__|____75.62_____|____82.13____
> _DiffPool-NOLP_ |___ 79.98__|____76.22_____|____75.58____
> ______DiffPool____|___ 80.64__|____76.25_____|____75.48____
> ______g-U-Net____|___ 82.43__|____77.68_____|____77.56____
>
> Due to time constraint, we will add these results in the final version of our paper.
>
> "-In a related point, while this work cites other approaches that apply pooling operations in graph neural networks (e.g., Ying et al. 2018, Simonovsky and Komodakis 2018), no comparisons are made against these approaches... "
>
> The work proposed in DiffPool can be used for constructing un-pool layers. Our proposed approaches are similar to regular pooling and un-pooling layers used on images and texts. We selected some important nodes to form a new graph using original edges. For DiffPool, the graph will become softly connected with every two nodes connected by a probability rate. In addition, our proposed pooling layer only involves a very small number of extra parameters, which are trainable projection vectors. While in DiffPool, a network is employed for each diff-pool layer to learn the assignment matrix. That may increase the risk of overfitting and make the training unstable. Actually, to stabilize training, DiffPool employs an auxiliary task of link prediction during training in graph classification tasks.
>
> "-As another minor point, whereas unpooling operations can be used to define a generative model in the image setting, this is not the case here, as the unpooling operation relies on knowledge about the input graph (i.e., the model always unpools to the same connectivity structure). This is not necessarily a bad thing, but it could improve the paper to clarify this issue. "
>
> Sure, our proposed gUnpool layer corresponds to the regular un-pool layer used on images. The regular un-pool layer also needs the pooled position information in corresponding regular pooling layer to restore the original image structure. We will add this clarification in the final version of our paper.
>
> [1] Rex Ying, Jiaxuan You, Christopher Morris, Xiang Ren, William L Hamilton, and Jure
> Leskovec. Hierarchical graph representation learning with differentiable pooling. The Thirty-second Annual Conference on Neural Information Processing Systems (NIPS), 2018

---

> > ### Comment · AnonReviewer3 · 2018-11-21
> > **Thanks for the response!**
> >
> > Thanks for the thorough response! Adding the new graph classification results is great, and I agree that a strength of the proposed approach is that it is simpler and less prone to overfitting (e.g., it does not need the auxiliary link prediction objective for stability).
> >
> > I've revised my score to reflect your response.

---

> > > ### Author Response · Authors · 2018-11-21
> > > **Thank you**
> > >
> > > Thank you very much.

---

### Public Comment · (anonymous) · 2018-09-29
**tanh gating?**

Very interesting work!

I was wondering, why the hyperbolic tangent activation was used for the gating mechanism in your architecture? The choice doesn't seem to be motivated anywhere in the paper, and given that its output can be negative (and therefore inadvertently flip the activation), the logistic sigmoid should be more appropriate.

Could you please comment on this decision?

Thanks!

---

> ### Author Response · Authors · 2018-09-30
> **Why use tanh gating**
>
> Hi, thank you for your appreciation and question. Actually, we have tried sigmoid, tanh and softmax. tanh performs the best. We have thought about reasons. There are some possible explanations. The values in y vector are the scalar projection values. The negative values do not mean they are negligible just because they are in the opposite direction of vector p. So if we do sigmoid, their corresponding node vectors will become trivial. And also tanh is zero centered, which facilitates the training of projection vectors. So we choose to use tanh for gate operation. Also the use of tanh can regularize node vectors such that they are in the same direction of projection vector p. We are not sure if this can help with the feature encoding. We may try to investigate this in the future. Hope these explanations can help you. Happy to have future discussion with you if any question. Thank you.

---

> > ### Public Comment · (anonymous) · 2018-09-30
> > **possible further studies on gating**
> >
> > Thank you for the prompt reply!
> >
> > While I am surprised that the tanh performs the best, if this is indeed the case then of course you should use it. I would definitely recommend that you clarify (within the paper) how the tanh function was arrived at during the revision period.
> >
> > Further, assuming your intuition about the benefits of tanh is correct, then the vectors substantially opposite of p could be useful too, right? This motivates another experiment, where you’d take the top-k from y^2, rather than y (to give the opposite direction equal footing).
> >
> > What do you think?
> >
> > Once again, thanks for promptly responding to my query, and best of luck with the reviews.

---

> > > ### Author Response · Authors · 2018-09-30
> > > **possible further studies on gating**
> > >
> > > Sure, I totally agree with you. We can do more experiments about this part. Very happy to have this great discussion with you.

---

### Public Comment · ~Michael_Bronstein1 · 2018-09-30
**important baselines missing**

I believe that many important baseline algorithms for deep learning on graphs are missing, in particular CayleyNet [1] (a generalization of ChebNet using rational functions) MoNet [2] (a more general model of which GAT is a subsetting), and the recent Dual/Primal Graph CNN [3]. Please refer to a review paper [8] on geometric deep learning methods.

1. CayleyNets: Graph convolutional neural networks with complex rational spectral filters, arXiv:1705.07664,

2. Geometric deep learning on graphs and manifolds using mixture model CNNs, CVPR 2017.

3. Dual-Primal Graph Convolutional Networks, arXiv:1806.00770.

4. Geometric deep learning: going beyond Euclidean data, IEEE Signal Processing Magazine, 34(4):18-42, 2017

---

> ### Author Response · Authors · 2018-10-18
> **Baselines**
>
> Thank you for the interest in our work and references. We were aware of these work. But our work mainly focus on graph pooling and un-pooling operations, which are orthogonal to methods in these papers. We would like to add these references as needed in our final version.

---

### Public Comment · ~Jingjia_Huang1 · 2018-12-08
**What's about the sparsity of the graph after gPool?**

Intuitively, nodes that are close in the spatial domain and have similar neighbours will have similar node embeddings after the GCN layer, which means they would be likely to have close ranking scores. As a result,  the whole graph may be pooled into a certain group of nodes that are close in the graph, which probably destroys the inherent stucture of the original graph.
So, my question is did you do any work to visuaize the gPool progress? Are the selected nodes sparsely distributed  in different parts of the graph or stayed very close to each others?

---

### Public Comment · (anonymous) · 2018-12-15
**Interesting idea, but needs stronger experiments and some clarification**

I found the proposed idea interesting, but there are a few issues in the experiments that should be addressed.

1. Graph augmentation seems to be important to get state of the art results. Without it, this work is better than GAT only on one dataset (Cora) in node classification tasks. It is also not clear which step of graph augmentation is more important: using power 2 of adjacency matrix A or using weighted self connections 2I. To sum up, methods in Table 2 ideally should use the same preprocessing/augmentation, otherwise it is not a fair comparison.
2. For the graph classification experiments added as a comment here, it is hard to make any conclusion because of lack of details. For COLLAB the results in a couple of papers are better (80.7% in WL-OA [1] and, as mentioned by the authors, 82.13% in DiffPool [2]). Also, it is not clear how you cope with the fact the nodes are featureless in this dataset and node features are required for your model to learn the projection vector. Do the authors add artificial features such as node degrees or use any node embedding layer to generate strong features before feeding them to Graph U-Net similar to how they do in the node classification tasks? If so, these features/this layer should also be added to the baseline methods for fair comparison. For PROTEINS it is not clear whether authors used continuous node attributes in addition to discrete features. Previous works like WL-OA [1] use only discrete features. The results on D&D look very good and it would be great to report more results on large graphs for which efficient and fast pooling should prove beneficial. Some experiments with random large graphs would be a good start to show that it is much faster (?) than other pooling methods.
3. Have you tried adding node reconstruction loss for the graph classification task to improve the model?
4. I also agree with other reviewers that the proposed way to solve the problem of isolated nodes after pooling is not the best, yet the problem seems to be critical. Another challenging problem already touched by others is that some groups of nodes can be ignored as a result of pooling. To be convincing, the authors should somehow better show (quantitatively, qualitatively, theoretically, etc.) that they either solve these problems or that these problems are not a big deal. Again, authors could start with some synthetic graphs.

I was trying to reproduce the graph classification results as in the authors’ comments here, but so far results using Graph U-Net are worse than just using baseline GCN (Kipf & Welling, 2017).
Can the authors provide all necessary details to reproduce graph classification results?
Since hyperparameters for graph classification models are not provided and I guess they are different from hyperparameters of node classification models,  I am using hyperparameters of a related recent work [3] that adopted the pooling method from this submission.
My implementation is available at https://github.com/bknyaz/graph_nn.

Overall, I understand that the purpose of this paper is to show that U-Net like architecture is also great for graph structured data. But given experiments, I am not completely convinced to apply this model to some graph problem.

[1] Nils M. Kriege, Pierre-Louis Giscard, Richard C. Wilson. On Valid Optimal Assignment Kernels and Applications to Graph Classification.
[2] Rex Ying, Jiaxuan You, Christopher Morris, Xiang Ren, William L Hamilton, and Jure
Leskovec. Hierarchical graph representation learning with differentiable pooling. The Thirty-second Annual Conference on Neural Information Processing Systems (NIPS), 2018
[3] Cătălina Cangea, Petar Veličković, Nikola Jovanović, Thomas Kipf, Pietro Liò, Towards Sparse Hierarchical Graph Classifiers, NIPS Workshop on Relational Representation Learning (R2L), 2018

---

### Public Comment · (anonymous) · 2019-10-14
**Test procedure is not valid !**

if that github is official code of the paper;

https://github.com/HongyangGao/gunet

the test procedure is not valid. Instead of doing a blind test, they reported the maximum test set accuracy during the epoch. So the results are not test results, but validation result.

To make it more clear, the researcher needs to stop their algorithm regardless of checking test accuracy. In that point, some certain number of predefined epochs can be used. Or some part of train set might be assigned as validation and algorithm  needs to stop according to validation set. For instance, I check the valdiation result of PROTEIN dataset yes it is 79% as it reorted, but when I stop the algorithm according to predefined epoch, the test result become %73.

So if I did not miss some point in the code, this method accuracy on PROTEIN is not 79% but 73% which is not good at all.
if the researcher clarifies that point, I appreciate it.

---

### Meta-Review · Area_Chair1 · 2018-12-14
**difficult case**

**Confidence:** 4
**Recommendation:** Reject

**Metareview:**

The authors supplied an updated paper resolving the most important reviewer concerns after the deadline for revisions. In part, this was due to reviewers requesting new experiments that take substantial time to complete.

After discussion with the reviewers, I believe that if the revised manuscript had arrived earlier, then it should be accepted. Without the new results I would recommend rejecting since I believe the original submission lacked important experiments to justify the approach (inductive setting experiments are very useful).

The community has an interest in uniform application of the rules surrounding the revision process. It is not fair to other authors to consider revisions past the deadline and we do not want to encourage late revisions. Better to submit a finished piece of work initially and not assume it will be possible to use up a lot of reviewer time and fix during the review process.

We also don't want to encourage shoddy, rushed experimental work. However, the way we typically handle requests from reviewers that require a lot of work to complete is by rejecting papers and encouraging them to be resubmitted sometime in the future, typically to another similar conference.

Thus I am recommending rejecting this paper on policy grounds, not on the merits of the latest draft. I believe that we should base the decision on the state of the paper at the same deadline that applies to all other authors.

However, I am asking the program chairs to review this case since ultimately they will be the final arbiters of policy questions like this.